# Genome-Wide Identification of *WRKY* Transcription Factor Family in Chinese Rose and Response to Drought, Heat, and Salt Stress

**DOI:** 10.3390/genes15060800

**Published:** 2024-06-18

**Authors:** Xinyu Yan, Jiahui Zhao, Wei Huang, Cheng Liu, Xuan Hao, Chengye Gao, Minghua Deng, Jinfen Wen

**Affiliations:** 1Faculty of Architecture and City Planning, Kunming University of Science and Technology, Kunming 650021, China; yanxinyu20000812@163.com (X.Y.); zhaojiahui2213@163.com (J.Z.); liucheng102600@163.com (C.L.); haoxuan031300@163.com (X.H.); gaochengye0921@163.com (C.G.); 2College of Agronomy and Life Sciences, Kunming University, Kunming 650021, China; huangwei_kmu@163.com; 3Key Laboratory of Vegetable Biology of Yunnan Province, College of Landscape and Horticulture, Yunnan Agricultural University, Kunming 650201, China

**Keywords:** Chinese rose, *RcWRKY*, phylogenetic analysis, expression pattern

## Abstract

The *WRKY* gene family is a key transcription factor family for plant development and the stress response. However, few studies have investigated the *WRKY* gene family in Chinese rose (*Rosa chinensis*). In this study, 68 *RcWRKY* genes were identified from the Chinese rose genome and classified into three primary groups and five subgroups based on the structural and phylogenetic characteristics. The analysis of the conserved domains, motifs, and gene structure revealed that the *RcWRKY* genes within the same group had the same exon–intron organization and composition. Chromosome mapping and gene duplication revealed that the *RcWRKY* genes were randomly dispersed across seven chromosomes. Fragment duplication and refined selection may have influenced the evolution of the *WRKY* gene family in Chinese rose. The *cis*-acting elements in the *WRKY* promoter region revealed that the *RcWRKY* genes contained numerous abiotic stress response elements. The results of qRT-PCR revealed that the expression of *RcWRKY* was tissue-specific, with high expression being observed under drought, heat, and salt stress. Notably, *RcWRKY49*′s expression increased more than fivefold following salt stress, indicating that it is a crucial gene mediating the salt stress response of Chinese rose. These findings shed light on the regulatory role of *RcWRKY* in the growth and development of Chinese rose, and they serve as a foundation for future molecular breeding programs and gene discovery.

## 1. Introduction

Transcription factors (TFs) are crucial for plant growth and the stress response as they interact with *cis*-acting elements to regulate the transcription of downstream target genes. Since the discovery of the *WRKY* gene *SPF1* in sweet potato in 1994, researchers have identified the corresponding *WRKY* genes [1,2,3,4,5] from the three-flowered maple (*Acer triflorum*), chrysanthemum (*Dendranthema morifolium*), flax (*Linum usitatissimum*), and corn (*Zea mays*). Sixty to seventy amino acids constitute the WRKY family members’ conserved domain, which also includes a zinc finger structure (C2H2 or C2HC) and a highly conserved heptabenzo ring sequence (*WRKYGQK*) [6]. Based on the number of conserved domains and the type of zinc finger structure, the *WRKY* gene family members can be divided into three groups: I, II, and III. Group II is further divided into five subgroups: IIa, IIb, IIc, IId, and IIe [7,8]. Group I members have two conserved WRKY domains, while group II and III members have only one conserved WRKY domain. Group I and II members have a C-X4-5C-X23-24-H-X1-H (C2H2) zinc finger structure, while group III members have a C-X5-7-C-X23-38-H-X1-C (C2HC) zinc finger structure.

WRKY TFs can explicitly bind to the *cis*-acting elements in the promoter regions of target genes to regulate the expression of downstream genes and enhance plants’ resistance to low-temperature, drought, and salt stress, as well as pathogenic bacteria [7,8]. Although *GhWRKY21* has a negative regulatory effect on cotton (*Gossypium hirsutum*) under drought stress, inhibiting *GhWRKY21* expression can improve drought tolerance in cotton [9]. The overexpression of *MaWRKY80* in wild banana (*Musa acuminata*) improves the phenotypic traits, enhances drought tolerance, reduces the water loss rate, and reduces the reactive oxygen species levels in the leaves of transgenic *MaWRKY80* when compared to wild-type *Arabidopsis thaliana* under drought stress [10]. Through the mediation of jasmonic acid (JA) and salicylic acid (SA) signaling, *AtWRKY11* and *AtWRKY70* can enhance plants’ resistance to *Bacillus*. The overexpression of *AtWRKY11* and *AtWRKY70* can enhance *A. thaliana*’s tolerance to drought stress, as well as enhancing seed germination and root development [11,12]. Sorghum (*Sorghum bicolor*) WRKY TF *SbWRKY30* is predominantly expressed in the leaves and roots and is induced under drought stress. The drought tolerance of maize can be increased by *ZmWRKY79* through the abscisic corrosive (ABA) biosynthetic pathway [13]. The expression level of *ClWRKY20* increases under saltiness, drought, ABA, and SA treatments. During seed germination, *ClWRKY20* overexpression increases the sensitivity of transgenic *A. thaliana* to ABA at low temperatures and in salinity [14]. In addition, *WRKY*, which is associated with drought stress, has been identified in other crops, which demonstrates the importance of the *WRKY* genes in regulating responses and adaptation to drought stress [15,16,17,18].

Chinese rose (*R. chinensis*) is an evergreen or semi-evergreen low shrub of the Rosaceae family. In China, the plant species is largely distributed in mountainous areas in Hubei, Sichuan, and Gansu Provinces. Chinese rose has high adaptability and drought tolerance; however, the species thrives in microstrip acidic sandy loam soils with rich organic matter and good drainage. Chinese rose, which is native to China, prefers sunny, warm, and humid climates [19,20]. Furthermore, Chinese rose is a valuable gene resource for the resistance breeding of Rosaceae because of its superior biological characteristics, such as cold resistance, drought resistance, salt and alkali tolerance, and other key traits, such as yellow flowers with purple and red spots, leaves without stipules, and a well-developed root system. The effect of exogenous calcium on Chinese rose exposed to salt stress has been investigated [21]. However, the resistance capacity and diversity of Chinese rose at a molecular level have not been studied. The Chinese rose genome has been sequenced and assembled at the chromosomal level. Whole-genome and gene annotation files of the Chinese rose genome are publicly available in the Ensembl Plants database (http://plants.ensembl.org/Rosa_chinensis/Info/Index, accessed on 1 April 2024) [1]. In this study, the *WRKY* gene family in the Chinese rose genome was identified using bioinformatics, and their phylogenetic relationships, physicochemical properties, gene mapping, and collinearity were analyzed. Sixty-eight *cis*-acting genes in the *WRKY* promoter region were screened and their expression levels under different types of stress and plant tissue were analyzed. Furthermore, some potentially interacting WRKY proteins were predicted. This study forms a basis for further research on the potential mechanisms underlying the responses of *WRKY* genes to stress in Chinese rose and the identification of candidate genes for stress resistance.

## 2. Results

### 2.1. Identification and Chromosomal Localization of WRKY Genes in Chinese Rose

To identify all members of the histone family in Chinese rose, 72 *Arabidopsis AtWRKY* protein sequences with moderate phylogenetic relationships with roses were used as seed sequences to search the whole-genome protein sequences of Chinese rose using BLASTP. The hidden Markov model (HMM) file of the histone domain (PF03106) was downloaded from the Pfam protein family database. All protein sequences were aligned with these HMMs using TBtools. Incomplete and candidate sequences without corresponding domains were excluded, according to the Pfam, SMART, and CDD databases. A total of 68 *RcWRKY* genes were identified from the Chinese rose genome (Appendix A). The genes were named *RcWRKY1–RcWRKY68* based on their positions on the seven chromosomes of Chinese rose.

*RcWRKY*’s sequence length ranged from 78 to 1574 amino acids. The isoelectric point (pI) values ranged from 4.87 to 9.85. The molecular weights ranged from 13629.17 kDa to 178605.88 kDa. The RcWRKY protein was primarily located in the nucleus, with the exception of RcWRKY4, which was located in the peroxisome. *RcWRKY15*, *RcWRKY50*, *RcWRKY10*, *RcWRKY65*, *RcWRKY8*, *RcWRKY26*, and *RcWRKY47* were located in the cytoplasm.

The results showed that the 68 *RcWRKY* genes were unevenly distributed on seven chromosomes in Chinese rose, and most were in high-gene-density positions (Appendix A). Among them, chromosomes chr1 and chr2 had the highest numbers of *RcWRKY* genes (12), although they were not the longest chromosomes. Chromosomes chr4 and chr6 had the lowest numbers of *RcWRKY* genes (6). Gene aggregation was observed on chr1, chr3, chr4, chr5, chr6, and chr7. The uneven distribution of the genes could be the result of the uneven replication of chromosome fragments in Chinese rose.

### 2.2. Phylogenetic Analysis of WRKY Genes in Chinese Rose

A phylogenetic tree of the WRKY protein sequence in Chinese rose and *A. thaliana* (reference gene) is illustrated in Figure 1. The RcWRKY proteins were divided into three groups (I–III). Based on the *AtWRKY* gene family members, group I had the most members (42 members), group II had 20 members, and group III had the least members (six members).

### 2.3. Motifs, Domains, and Gene Structures of WRKY Genes in Chinese Rose

The motifs, domains, and gene structures of the *RcWRKY* genes revealed the evolution of the *WRKY* gene family in Chinese rose. According to the constructed simplified phylogenetic tree, 68 RcWRKY proteins were divided into three groups (Figure 2A). All genes identified had 1–8 introns. In addition, genes within the same branch of the phylogenetic tree had a similar structure and their coding sequences (CDS) had similar numbers of introns. The position and length of the introns in the groups did not vary significantly, but significant variations were observed between groups.

The analysis of the structural characteristics of the RcWRKY proteins revealed that 10 conserved motifs contained 21–50 amino acids (Figure 2B). Most RcWRKY protein family members within the same group or subgroup had comparative conservative motifs, although slight differences were observed between the groups. Members of the same group or subgroup had a particular conservative motif. For instance, motif 3 only existed in group I members, and motifs 5 and 9 only existed in group II and III members. Based on these findings, it appears that gene evolution and function depend on the conserved motifs. In addition, almost all members of the RcWRKY family consisted of a WRKYGQK heptapeptide sequence (motif 1), indicating that WRKYGQK is the key conserved motif of the *RcWRKY* gene family.

A domain can be formed by several motifs. A protein molecule transcribed by a gene can contain multiple structurally specific and functionally different regions, which are called domains. The conserved domain names detected were the WRKYRPW8 superfamily, zf-CCCHAAAPLN03210 superfamily, TIRPlan-zn-clustWRKY superfamily, and C-JIDPRK13923 superfamily (Figure 2C).

Based on the results of gene annotation, the exon–intron coding sequence structure of the *RcWRKY* gene is as shown in Figure 2D. The results showed that most of the *RcWRKY* gene family members had 1–8 exons, except for a few members without untranslated regions (*RcWRKY47*, *RcWRKY4*, *RcWRKY60*, *RcWRKY15*, *RcWRKY17*, *RcWRKY11*, *RcWRKY53*, *RcWRKY58*, *RcWRKY40*, *RcWRKY23*, *RcWRKY64*, and *RcWRKY32*). Most of the *RcWRKY* genes contained two exons (three introns), accounting for 32.35% (22/68) of all *RcWRKY* genes. *RcWRKY* members within the same group or subgroup had similar gene structures and high conservation, which suggests functional similarities among the members.

### 2.4. Cis-Acting Elements of the WRKY Gene Family in Chinese Rose

The 2000 bp promoter sequence of the *RcWRKY* gene was extracted and its *cis*-elements were analyzed using the PlantCARE database to investigate the likely function of the *RcWRKY* gene in the abiotic stress response (Figure 3). A total of 31 *cis*-acting elements related to plant hormones and stress responses were identified in the promoter region of the *RcWRKY* gene, including light-, stress-, and growth and development-related response elements. The *cis*-acting elements related to the stress response included drought stress-, injury-, and hypoxia stress-related response elements. *RcWRKYs*, *RcWRKY5*, *RcWRKY14*, *RcWRKY16*, *RcWRKY28*, *RcWRKY39*, *RcWRKY49*, *RcWRKY56*, and *RcWRKY62* had the most *cis*-acting elements related to the stress response.

### 2.5. Collinearity Analysis of WRKY Genes in Chinese Rose

A total of 23 pairs of replicated genes were identified in the *RcWRKY* gene family (Figure 4 and Table 1). The 23 pairs of genes were defined as intrasegment and interchromosomal replications. Nine of the *RcWRKY* genes underwent multiple replications (*RcWRKY40*, *RcWRKY41*, *RcWRKY45*, *RcWRKY37*, *RcWRKY24*, *RcWRKY23*, *RcWRKY3*, *RcWRKY58*, and *RcWRKY60*). The nonsynonymous (KA) and synonymous (KS) substitution rates and KA/KS ratios demonstrating the evolution of the RcWRKY protein coding sequences are presented in Table 1. The KA/KS ratios of 19 replicated genes were less than one, indicating that the genes mainly underwent purified selection.

### 2.6. Collinearity Analysis of WRKY Genes in Chinese Rose and Other Species

The genome comparisons and collinearity analysis of the *WRKY* genes in four typical plant species (*A. thaliana*, *Ipomoea triloba* L., *Papaver somniferum* L., *Prunus persica* L.) and Chinese rose revealed 57, 95, 67, and 91 pairs of repeats between Chinese rose and *A. thaliana*, *Ipomoea triloba* L., *P. somniferum* L., and *P. persica* L., respectively (Figure 5). Several collinearities were detected between Chinese rose and plant species in the Convolvulaceae and Rosaceae families, suggesting that they are closely related.

### 2.7. Protein–Protein Interaction Network Analysis of WRKY Genes in Chinese Rose

The protein–protein interaction network analysis showed that the RcWRKY proteins with high sequence similarity to *AtWRKY40* (*RcWRKY54*), *AtWRK33* (*RcWRKY29*), *AtWRKY18* (*RcWRKY54*), and *AtWRKY70* (*RcWRKY61*) were the core nodes of the protein–protein interaction network. The rest of the RcWRKY proteins had strong or weak interactions with RcWRKY54, RcWRKY29, RcWRKY54, and RcWRKY61 (Figure 6).

### 2.8. Expression Patterns of WRKY Genes in Different Tissue Types of Chinese Rose under Drought, Heat, and Salt Stress

The expression levels of the *RcWRKY* genes in the roots, stems, and flowers were determined. The results showed that the expression levels of eight *RcWRKY* genes varied, suggesting that they perform different functions in plant development. The expression levels of *RcWRKY5* in the stems were the highest; the expression levels of *RcWRKY14*, *RcWRKY16*, *RcWRKY39*, and *RcWRKY49* in the roots were the highest; the expression levels of *RcWRKY14*, *RcWRKY39*, and *RcWRKY49* in the flowers were the lowest; the expression level of *RcWRKY16* in the leaves was the lowest; and the expression levels of other members were not significant (Figure 7).

The analysis of the expression of the *RcWRKY* genes under abiotic stress revealed that *RcWRKY5*, *RcWRKY14*, *RcWRKY16*, and *RcWRKY49* were expressed after 3 h of drought stress treatment, although their expression levels increased gradually with an increase in the treatment time. Similarly, the expression level of *RcWRKY39* increased with an increase in the treatment time. The expression levels of other members were not significant. *RcWRKY14* was significantly expressed after 9 h of exposure to heat stress, while *RcWRKY16*, *RcWRKY39*, and *RcWRKY49* were significantly expressed after 24 h of heat stress treatment. The expression of *RcWRKY5* and other members did not vary significantly with the prolongation of the treatment time. *RcWRKY14*, *RcWRKY16*, and *RcWRKY49* were significantly expressed after 3 h of salt stress treatment. No significant differences were observed in the expression of *RcWRKY5* and *RcWRKY39* with the prolongation of the treatment time. In general, the expression of *RcWRKY5*, *RcWRKY14*, *RcWRKY16*, *RcWRKY39*, and *RcWRKY49* was induced in Chinese rose under drought, heat, and salt stress conditions. *RcWRKY* exhibited a strong response to salt, drought, and heat stress. The expression level of *RcWRKY49* increased more than fivefold after exposure to salt stress, suggesting that *RcWRKY49* is a key gene associated with the response to salt stress (Figure 8).

## 3. Materials and Methods

### 3.1. Genome-Wide Identification and Analysis of Physicochemical Properties of Chinese Rose WRKY Gene Family

The whole-genome and gene annotation files of the Chinese rose genome were obtained from the Ensembl Plants database (http://plants.ensembl.org/Rosa_chinensis/Info/Index, accessed on 1 April 2024) [1]. The identification of the *WRKY* gene family in Chinese rose was performed as follows. First, the full-length amino acid sequences of AtWRKY4 (AT1G13960.1) in *A. thaliana* (http://www.arabidopsis.org/, accessed on 1 April 2024) were used as query sequences. The protein sequences were used as a template, while the protein sequences of the *WRKY* gene family in Chinese rose were preliminarily screened using BLAST in TBtools (v2.085), and an E-value of e−6 was used as the threshold [1]. Thereafter, the hidden Markov model (HMM) file of the WRKY domain (PF03106) was downloaded from the Pfam protein family database (http://pfam.sanger.ac.uk/, accessed on 1 April 2024) and the *WRKY* genes in the Chinese rose genome database were identified using TBtools [5]. Subsequently, the Pfam (http://pfam.xfam.org/search # tabview=Table 1, accessed on 1 April 2024), SMART (http://art.embl.de/, accessed on 1 April 2024), and NCBI CDD (https://www.ncbi.nlm.nih.gov/cdd/, accessed on 1 April 2024) databases were used to verify whether the RcWRKY proteins had a conserved WRKY domain. Candidate protein sequences without conserved domains were deleted. The physicochemical properties of the RcWRKY proteins, including the molecular weight, isoelectric point, and hydrophilicity, were analyzed using TBtools, with the parameters set to the default values. WoLF PSORT (https://wolfpsort.hgc.jp/, accessed on 1 April 2024) was used to predict the subcellular localization of the *RcWRKY* gene family members [22].

### 3.2. Phylogenetic Analysis and Chromosome Mapping of the WRKY Gene Family in Chinese Rose

The whole genome, protein sequences, and annotation file of the *WRKY* gene family were downloaded from the *Arabidopsis* Information Resource (TAIR) database (https://www.arabidopsis.org/, accessed on 1 April 2024) [23]. Multisequence alignment of the *AtWRKY* gene family proteins in Chinese rose and *A. thaliana* was carried out using DNAMAN 9.0. A phylogenetic tree was constructed based on the nearest neighbor-joining (NJ) method using MEGA 7.0, and 1000 bootstrap repetitions were performed. A staining map of the *RcWRKY* gene family was generated using TBtools based on the gene annotation file of the Chinese rose genome [24].

### 3.3. Gene Structure, Promoter, and Collinearity Analysis of WRKY Gene Family in Chinese Rose

The intron and exon features of the *RcWRKY* genes were analyzed using TBtools based on the gene annotation information [25]. The conserved motifs of the RcWRKY proteins were identified using MEME (http://meme.sdsc.edu/meme/, accessed on 3 April 2024) [26]. Peptide sequences for the *RcWRKY* proteins were searched using Phytozome 13.0, and the conserved domains in the protein structure were identified using the NCBI CDD database. Motif widths of up to 10 motifs and 6–150 amino acid residues, as well as the gene structure, conserved domains, and motifs, were visualized using TBtools based on arbitrary repetition. Promoter sequences of up to 2000 base pairs in the upstream CDS regions of the *RcWRKY* genes and *cis*-acting elements were obtained using TBtools [27]. The light-responsive elements of the *RcWRKY* genes were predicted using the PlantCARE database (http://bioinformatics.psb.ugent.be/webtools/plancare/html/, accessed on 6 April 2024). The collinearity data of the *RcWRKY* genes were obtained using MCScanX, and fragment and tandem gene replications were identified based on the Chinese rose genome data [28]. The following criteria were used to determine gene duplication events: assuming that two genes were located in the same chromosome region and were adjacent or separated by one gene, the genes were considered to be tandem replications [29]. TBtools was used to calculate the KA, KS, and KA/KS values of the *RcWRKY* genes.

### 3.4. Protein–Protein Interaction Network Analysis of WRKY Genes in Chinese Rose

The RcWRKY protein–protein association network was built using the Search Tool for the Retrieval of Interacting Genes/Proteins (STRING; https://string-db.org/, accessed on 10 April 2024) database, with the *Arabidopsis* WRKY protein as a template. Five interactions were identified based on the protein–protein interaction score (≥0.700) [30]. A strong interaction was indicated by a thick line between the interaction network targets. Cytoscape (v3.10.0) was used to import the results and visualize the protein–protein interaction networks [31].

### 3.5. Plant Material, Stress Treatments, and Tissue Collection

In this study, the Chinese rose cultivar “*R. chinensis* Jacq.” was cultivated under an ambient temperature of 25 °C and 16 h/8 h light/dark conditions for 14 days. Samples (0.2 g) were collected from the roots, stems, leaves, and petals of the Chinese rose plants. The samples were immediately frozen in liquid nitrogen and stored at −80 °C until RNA extraction.

For the drought, heat, and salt stress treatments, Chinese rose cuttings of uniform size were soaked in quarter-strength Hoagland’s solution and 20% (*w*/*v*) polyethylene glycol (PEG) 6000 solution (pH 6.5). The plants were divided into four groups: (1) plants treated with quarter-strength Hoagland’s solution (CK); (2) plants treated with 20% PEG 6000; (3) plants treated with quarter-strength Hoagland’s solution and incubated under light and heat (42 °C) conditions; (4) plants treated with quarter-strength Hoagland’s solution and 200 mmol/L NaCl. Samples of leaves (the first leaf under the flower) were collected at 0, 3, 9, and 24 h after exposure to the stress treatments. Afterward, the samples were immediately frozen in liquid nitrogen and stored in a refrigerator at −80 °C until RNA extraction. Three repetitions per treatment were used for analysis.

### 3.6. Analysis of WRKY Gene Expression in Chinese Rose under Drought Stress by qRT-PCR

RNA was extracted using the TRIzol reagent according to the methods described by Yin et al. (2008) [32] and Chi et al. (2021) [33]. Reverse transcription was performed using the RevertAid First-Strand cDNA Synthesis Kit (K1622; Thermo Fisher Scientific, Waltham, MA, USA), according to the manufacturer’s instructions. The first-strand cDNA obtained by reverse transcription was packaged and stored at −80 °C until analysis. The sequences were extracted based on the whole-genome sequence of Chinese rose. The specific primers were designed with Primer 5.0 (Appendix A). The primers were synthesized by the Kunming Branch of Beijing Qingke Biological Co., Ltd. (Kunming, China). The reaction system consisted of Hieff qPCR SYBR Green Master Mix (Shanghai Yisheng Biotechnology Co., Ltd., Shanghai, China), 10 µL; positive and negative primers, 0.4 µL each; template DNA, 2 µL; and sterile ultrapure water, 7.2 µL. The amplification procedure consisted of 95 °C pre-denaturation for 5 min, 95 °C denaturation for 10 s, 60 °C annealing for 30 s, and extension at 72 °C for 30 s and 40 cycles. The qPCR analysis was performed with *Rcactin* as the internal reference gene. The relative gene expression was calculated using the 2−∆∆CT method.

## 4. Discussion

The *WRKY* TF family is a type of plant-specific supergene family that plays a key role in plant growth and development, as well as biotic and abiotic stress. The numbers and types of *WRKY* genes in different plant species vary. For example, there are 74 *WRKY* genes in *A. thaliana*, 103 in rice (*Oryza sativa*), 64 in strawberry (*Fragaria vesca*), 54 in maple (*A. triflorum*), 138 in citrus (*Citrus reticulata*), 102 in flax (*L. usitatissimum*), and 125 in corn (*Z. mays*). These variations could be associated with the genetic background as well as the growth and evolutionary histories of plant species [34]. In this study, 68 *RcWRKY* genes distributed on seven chromosomes were identified from the Chinese rose genome. In addition, several genes were clustered in the same chromosome segment to form a gene cluster, indicating that these genes share biological functions due to the fact that they are regulated and co-expressed together. The *WRKY* gene family members can be divided into three groups, groups I, II, and III, based on the number of conserved domains and the type of zinc finger structure. Group II is further divided into five subgroups: IIa, IIb, IIc, IId, and IIe [6]. Sixty-eight *RcWRKY* genes were divided into three major groups and five subgroups in this study, which was consistent with previous findings. Group I consisted of 42 WRKY proteins, indicating that the group could be the main driving force for the expansion of the *WRKY* gene family. Group III consisted of only six WRKY proteins, which could be due to its shorter evolutionary history compared to the other groups [35].

Gene structures can provide essential information for an understanding of the evolution of a gene family. The *WRKY* gene structures in different plant species vary [36]. Variations in the number and distribution of exons and introns of the WRKY genes in different plant species have also been observed. For example, the *MdWRKY* gene in apple (*Malus domestica*) is highly conserved in structure and most of the genes consist of 2–5 exons [37]. Most *CtWRKY* genes in safflower (*Carthamus tinctorius*) consist of four exons (two introns) [38]. Most *RcWRKY* genes identified in this study consisted of two exons (three introns). The difference could be associated with the genetic background, evolution, and adaptability of the species. For example, the *WRKY* genes of some plant species may have more exons to encode more complex protein structures that enhance their adaptation to complex changes. In other plant species, the *WRKY* genes have fewer introns to enhance the efficiency of gene expression due to the complexity of transcription and splicing [39]. As key regulators of plant gene expression, WRKY TFs usually have a typical gene structure that includes exons and introns. However, a few members of the WRKY gene family in Chinese rose have no exons, which is consistent with observations of the *WRKY* gene family in safflower [38]. The possible reasons for these observations are as follows. First, exons may be atypical gene structures. Although most *WRKY* genes have a typical exon–intron structure, some types of *WRKY* genes have structures that differ from those of conventional *WRKY* genes. However, this does not imply that such genes have no exons; their structures may have been altered to a certain extent. Second, the observations could be attributed to unique evolutionary mechanisms. In some cases, gene structures can be altered through unique evolutionary mechanisms, such as intron loss [40].

The gene replication events of the *WRKY* family are important in the evolution of plant genomes because they lead to diversity among the members and functions of the *WRKY* family. The gene replication events of the *WRKY* family have characteristics that are specific to different plant species. For example, 89 *WRKY* genes have been identified in pea (*Pisum sativum*) and these genes have been shown to be randomly distributed on chromosomes, with nine pairs of tandem repeats and 19 pairs of segment repeats [41]. A total of 68 *CtWRKY* genes have been identified among the *CtWRKY* TF genes, including five tandem-replicated gene pairs and 34 fragment-replicated gene pairs, which collectively constitute 82.93% of the *WRKY* family members in *CtWRKY.* These replication events are crucial for the evolution of the *WRKY* gene family in safflower [38]. In this study, 23 fragmented and duplicated gene pairs were identified among the *RcWRKY* TF genes, including 46 *RcWRKY* genes, which accounted for 67.65% of the *WRKY* family members in Chinese rose. This finding suggests that the fragment duplication events of the *WRKY* gene family are key factors influencing evolution in Chinese rose when compared to pea and safflower. Chinese rose is a diploid; therefore, the WRKY genes are not only subgenomic homologous pairs. These events are of great significance for plants’ adaptation to complex and challenging environmental conditions based on the number and functional diversity of genes. Replication events not only increase the number of *WRKY* genes, but also produce *WRKY* genes with novel functions or optimize existing functions through subsequent mutation and selection. Gene replication is also closely associated with the evolutionary processes and adaptive mechanisms of plants [42]. For example, the *WRKY* gene can produce more members with specific response functions through replication and differentiation in plant responses to biotic and abiotic stress. These members are associated with different transduction pathways that regulate plant growth and development and defense responses, thereby improving plants’ adaptability and viability [43]. Notably, not all replicated genes can be preserved, although the replication of the *WRKY* gene family results in diversity in its members and functions. Some replicated genes may be eliminated due to functional redundancies or harmful mutations, while others may be retained by natural selection to perform new functions. Therefore, multiple factors, including the occurrence, retention mechanism, and functional changes associated with *WRKY* gene replication events, should be taken into consideration when studying the gene replication events of the *WRKY* family. Overall, the gene replication events of the *WRKY* gene family are an important component of the plant genome’s evolution [44]. However, the specific mechanisms and influence of WRKY gene family replication require further research.

Concerning the interactions of WRKY proteins in other species, the potential gene-regulatory function of the RcWRKY protein and its relationship with biological functions can be predicted. In this study, protein–protein interaction network analysis uncovered that three RcWRKY proteins (RcWRKY54, RcWRKY29, and RcWRKY61) had high sequence similarity to AtWRKY40, AtWRKY33, AtWRKY18, and AtWRKY70; these proteins were distinguished as core nodes in the interaction network. These proteins interact with other proteins to varying degrees. AtWRKY40 can form a dimer with AtWRKY18 or AtWRKY60 to regulate osmotic adjustment, increase plant pathogen resistance, boost the antioxidant capacity, and improve drought tolerance [45]. AtWRKY33 can render *A. thaliana* more resistant to low temperatures, salt stress, and pathogenic fungi [46,47]. In *A. thaliana*, AtWRKY53 regulates drought tolerance by modulating stomatal movement; this is accomplished by either decreasing the hydrogen peroxide content or increasing starch synthesis [48,49]. Homologous proteins with similar domains and sequences in different species may have the same functions. Therefore, RcWRKY54, RcWRKY29, and RcWRKY61, in addition to other proteins, could be involved in the transcriptional regulation of the stress response in Chinese rose. Overall, the strong and weak interactions observed between the RcWRKY proteins suggest that the proteins are co-expressed and they regulate the resistance of Chinese rose to biotic and abiotic stress.

*Cis*-acting elements are crucial “switches” for the regulation of gene expression in plants. Numerous abiotic stress-responsive *cis*-acting elements have been identified in the promoter regions of the *WRKY* gene family in several plant species, indicating that the *WRKY* genes regulate the transcriptional response to abiotic stress in most plants [25]. In this study, 31 *cis*-acting elements related to the plant stress response were identified in the promoter regions of the *RcWRKY* genes. The 31 response elements included light-, stress-, and plant growth and development-related response elements. The *cis*-acting elements related to the stress response included drought stress, injury, and hypoxia stress response elements. Previous studies have shown that *WRKY* TFs regulate the responses to abiotic stress by binding to transcriptional regulatory elements related to stress. For example, the *ZmWRKY65* gene in maize can be transcriptionally activated by a variety of factors, including drought-, salinity-, heat-, and plant defense-related hormones, such as SA and ABA, to improve its resilience against abiotic stress [50]. The expression of the *McWRKY57* gene is induced by mannitol, ABA, and methyl jasmonate. The overexpression of the *McWRKY57*-like gene in *A. thaliana* considerably improves the drought tolerance capacity of plants [51]. Furthermore, the overexpression of *EjWRKY17* in transgenic *A. thaliana* has been reported to promote cotyledon greening and root lengthening under ABA treatment [52].

The functions of *cis*-acting elements in response to stress can be evaluated by comparing the expression levels of genes under different treatment conditions. To date, numerous studies have investigated the expression of WRKY TFs in other plant species under stress. For example, the expression level of *TaWRKY*s in wheat (*Triticum aestivum*) increases rapidly under drought stress, suggesting its crucial role in regulating the wheat drought response [53]. The rapid response facilitates growth and increases the wheat yield under drought stress. The overexpression of *PbrWRKY53* in birchleaf pear (*Pyrus betulifolia*) increases drought resistance [54]. *PbrWRKY53* enhances the adaptability of birchleaf pear to drought by regulating the antioxidant capacity and *PbrNCED1* expression. WRKY TFs are associated with the tomato seedling response to low-temperature stress. These TFs are activated at low temperatures and they enhance the adaptability of tomato seedlings to low temperatures by regulating the expression of downstream genes. The overexpression of WRKY TFs in rice improves its cold resistance capacity by activating the expression of antioxidant enzyme and lipid transport genes to protect the plasma membrane from damage caused by cold [55]. OsWRKY71 positively regulates cold tolerance in rice by regulating downstream target genes [56]. In this study, *RcWRKY5*, *RcWRKY14*, *RcWRKY16*, *RcWRKY28*, *RcWRKY39*, *RcWRKY49*, *RcWRKY56*, and *RcWRKY62* had the most *cis*-acting elements related to the stress response. Therefore, these eight genes were selected to investigate the stress responses of the *RcWRKY* genes under drought, heat, and salt stress. Our results revealed that the expression levels of *RcWRKY9* and *RcWRKY55* were more than fivefold higher than those of the control group under salt stress, suggesting that they are involved in enhancing salt stress tolerance. However, the potential of these genes as key regulatory factors of salt stress and the mechanisms underlying this phenomenon require further study. Based on the results of studies that have investigated the functions of the *WRKY* gene family in several plant species, we can speculate that there are more *RcWRKY* genes that are related to abiotic stress and could have unknown regulatory functions.

## 5. Conclusions

Chinese rose is an important ornamental flower with medicinal and nutritional values. A total of 68 *RcWRKY* genes were identified from the Chinese rose genome in this study. Bioinformatics and gene expression analyses revealed the key functions of the *RcWRKY* genes in plant growth and development and the abiotic stress response. Eight *RcWRKY* genes were responsive to drought, heat, and salt stress based on qRT-PCR analysis, indicating that they had positive and negative regulatory effects on the response of Chinese rose to the three types of abiotic stress. In addition, *RcWRKY14* and *RcWRKY16* were associated with rapid responses to drought, heat, and salt stress, which provides a basis for further studies on candidate genes related to stress resistance in Chinese rose. The results of this study form a theoretical basis for the further exploration of the *RcWRKY* genes’ functions in growth and development and the response to abiotic stress in Chinese rose. This study also provides a basis for the breeding of new varieties with excellent genetic characteristics, such as drought resistance, salt tolerance, and heat resistance, which would be conducive to agricultural development.

## Figures and Tables

**Figure 1 genes-15-00800-f001:**
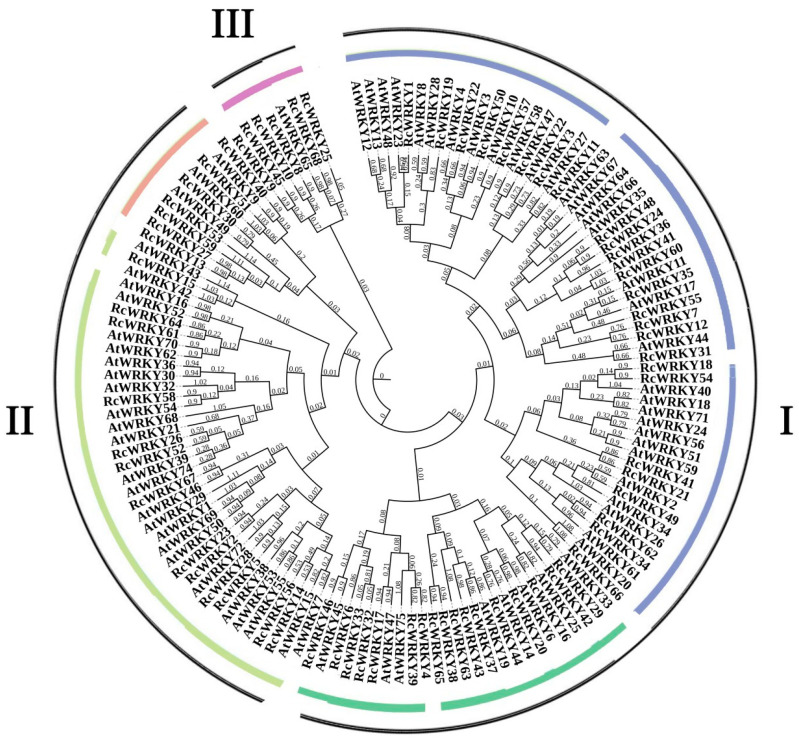
Phylogenetic analysis of Chinese rose and Arabidopsis WRKY proteins. An unrooted neighbor-joining (NJ) phylogenetic tree was constructed with the WRKY domains of the WRKY proteins from Chinese rose and Arabidopsis using MEGA7.0 with a bootstrap of 1000. Three major groups are indicated: I, II, and III. Subgroups are represented by different colored areas.

**Figure 2 genes-15-00800-f002:**
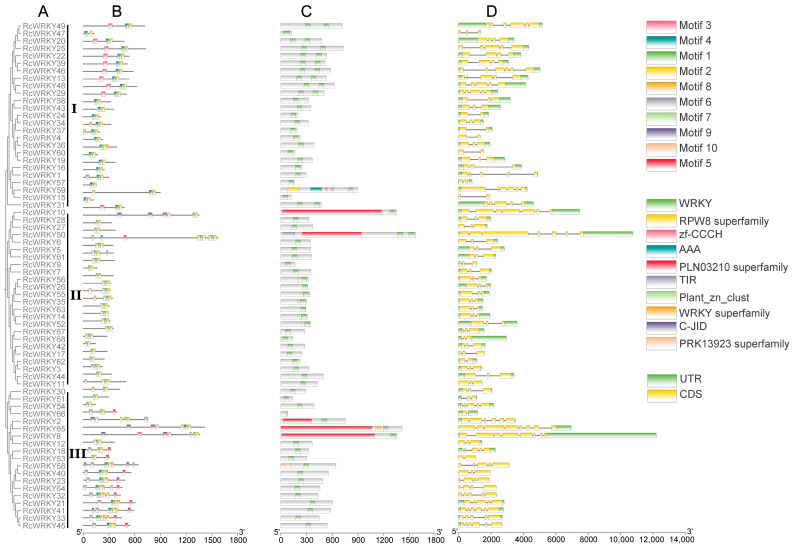
Phylogenetic tree, motif, domain, and gene structure of RcWRKY in Chinese rose: (**A**) phylogenetic tree of RcWRKY protein; (**B**) motif composition of RcWRKY; (**C**) domain composition of RcWRKY; and (**D**) gene structure of RcWRKY gene in Chinese rose. The phylogenetic tree was constructed from the RcWRKY protein sequences using the MEGA 7.0 software with the neighbor-joining method. The conserved motifs and domains of the RcWRKY proteins were analyzed using the MEME tool. The structures of all 68 RcWRKY genes were obtained by the TBtools software.

**Figure 3 genes-15-00800-f003:**
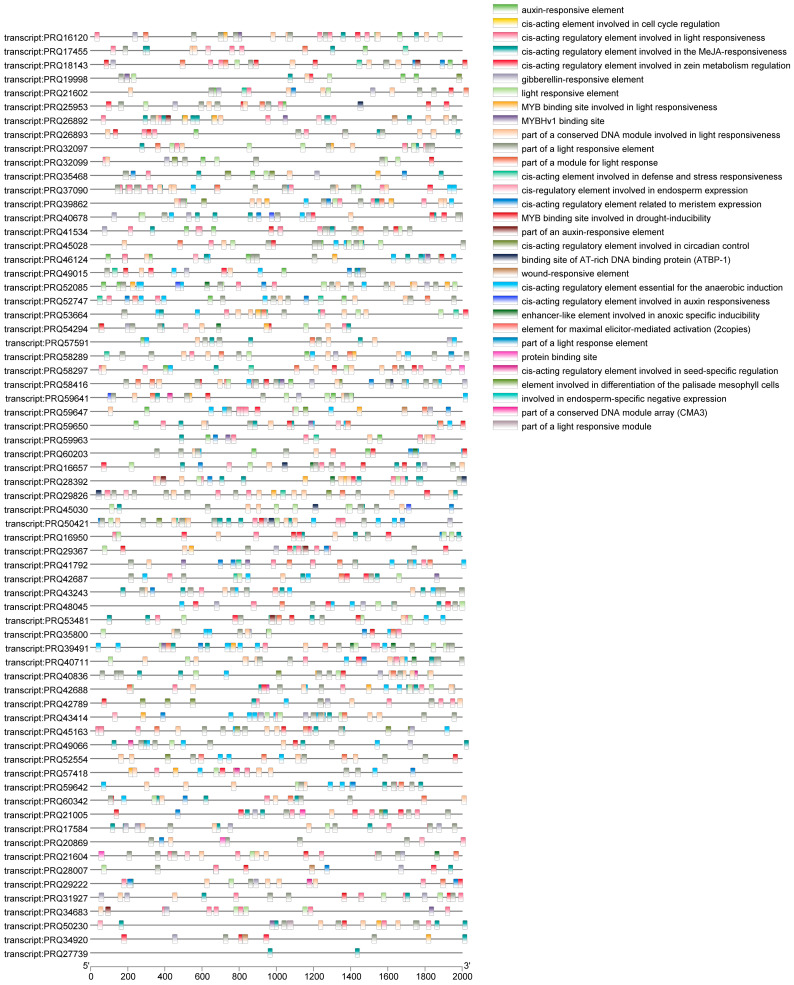
Distribution of *cis*-acting elements in the promoter region of the WRKY gene in Chinese rose.

**Figure 4 genes-15-00800-f004:**
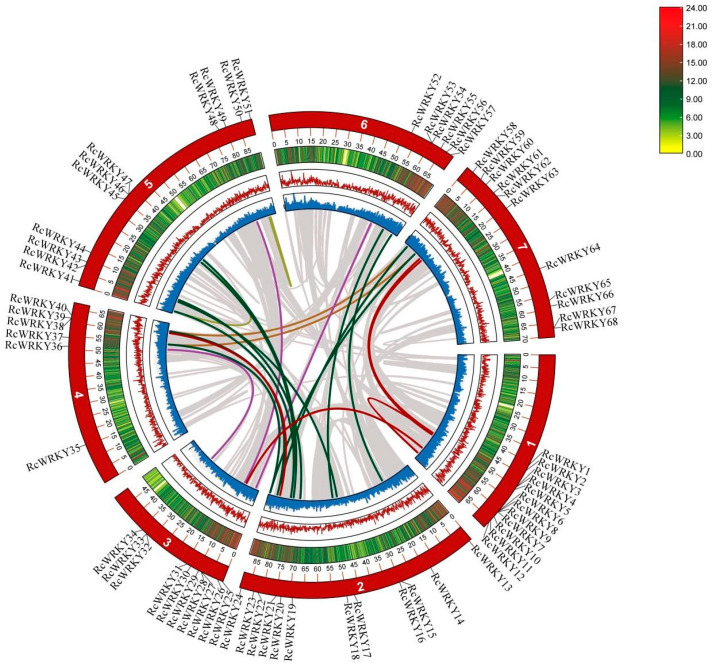
Collinear analysis of WRKY genes in Chinese rose; colored lines represent duplicated gene pairs in RcWRKY and gray lines represent collinear gene pairs in the genome.

**Figure 5 genes-15-00800-f005:**
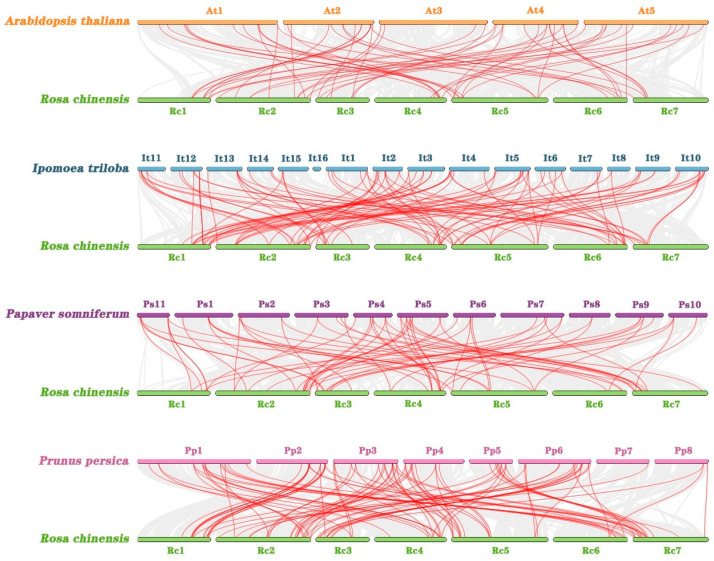
Collinearity analysis of WRKY genes in in Chinese rose and four representative plants. All RcWRKY homologous genes in *R. chinensis*, Arabidopsis thaliana, Ipomoea triloba, Prunus persica, and Prunus persica are denoted by red lines. Collinear blocks of Chinese rose and other species are shown as gray lines.

**Figure 6 genes-15-00800-f006:**
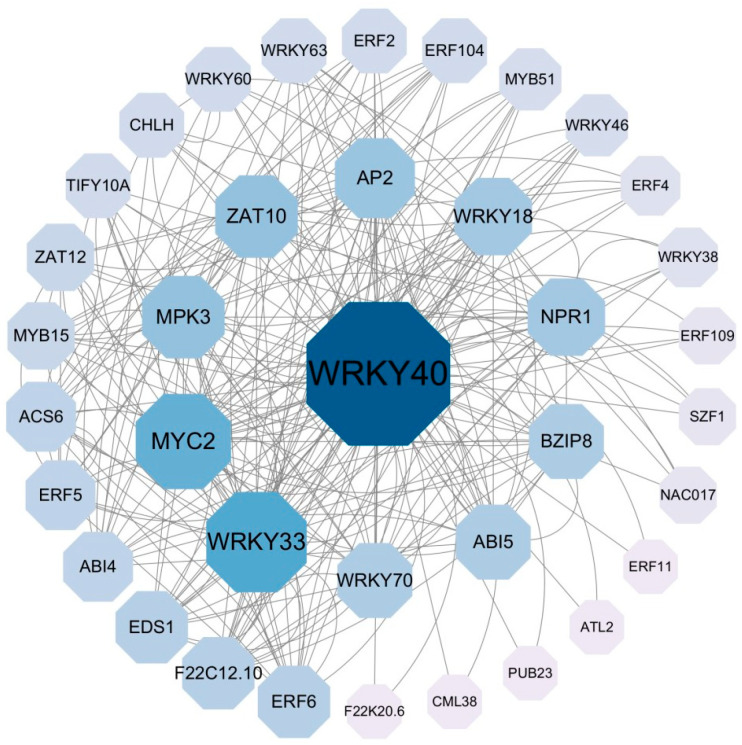
Chinese rose WRKY protein interaction network. The network was visualized using the Cytoscape software. The blue ball (node) represents the RcWRKY gene. Connecting lines indicate potential regulatory relationships. Text shows predicted gene names.

**Figure 7 genes-15-00800-f007:**
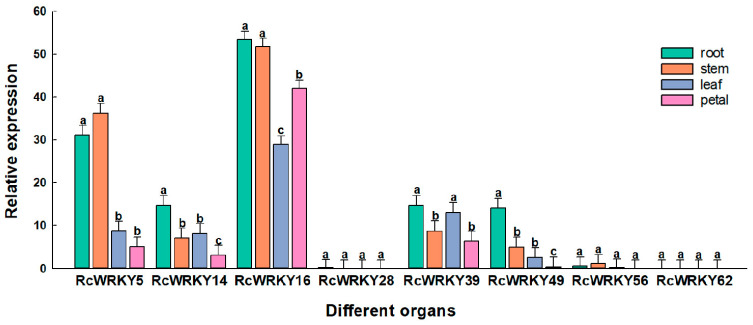
Relative expression levels of the WRKY gene in Chinese rose in different tissue types. Rcactin is quantified as an internal control. X-axes represent various treatments (0 h, normal conditions; 3 h, 9 h, and 24 h indicate hours of stress treatment), and y-axes are scales of relative expression levels. Data represent means ± SEs of three reproducible experiments. Bars with lowercase letters above the columns indicate significant differences between the four development periods at *p* < 0.05 (Tukey’s test).

**Figure 8 genes-15-00800-f008:**
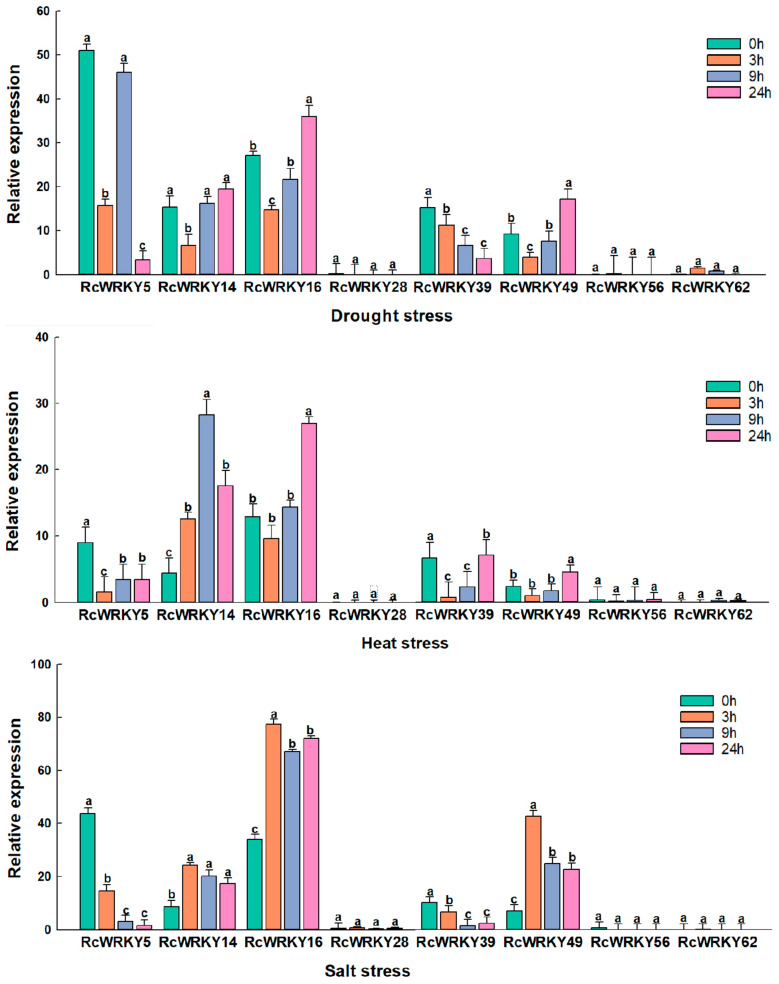
Relative expression levels of the WRKY gene in Chinese rose under drought, heat, and salt treatment. Rcactin is quantified as an internal control. X-axes represent various treatments (0 h, normal conditions; 3 h, 9 h, and 24 h indicate hours of stress treatment), and y-axes are scales of relative expression levels. Data represent means ± SEs of three reproducible experiments. Bars with lowercase letters above the columns indicate significant differences between the four development periods at *p* < 0.05 (Tukey’s test).

**Table 1 genes-15-00800-t001:** Intrasegment replication of WRKY homologous pairs in Chinese rose.

Paralogous WRKY Pairs	Chr Location	Duplication Type	Ka	Ks	Ka/Ks
RcWRKY-3	Chr1	Segmental	0.5702	2.4233	0.2353
RcWRKY-11	Chr1				
RcWRKY-7	Chr1	Segmental	0.6036	2.4398	0.2474
RcWRKY-27	Chr3				
RcWRKY-5	Chr1	Segmental	0.4942	2.2465	0.2200
RcWRKY-61	Chr7				
RcWRKY-3	Chr1	Segmental	0.5012	1.5593	0.3214
RcWRKY-62	Chr7				
RcWRKY-22	Chr2	Segmental	0.3623	2.5665	0.1412
RcWRKY-39	Chr4				
RcWRKY-23	Chr2	Segmental	0.4049	1.9961	0.2029
RcWRKY-40	Chr4				
RcWRKY-24	Chr2	Segmental	0.3613	2.5494	0.1417
RcWRKY-37	Chr4				
RcWRKY-21	Chr2	Segmental	0.2585	1.8568	0.1392
RcWRKY-41	Chr5				
RcWRKY-19	Chr2	Segmental	0.4138	2.4782	0.1670
RcWRKY-43	Chr5				
RcWRKY-17	Chr2	Segmental	0.4729	NaN	NaN
RcWRKY-42	Chr5				
RcWRKY-21	Chr2	Segmental	0.3524	1.6203	0.2175
RcWRKY-45	Chr5				
RcWRKY-20	Chr2	Segmental	0.5243	1.2481	0.4201
RcWRKY-46	Chr5				
RcWRKY-14	Chr2	Segmental	0.3136	4.7404	0.0661
RcWRKY-56	Chr6				
RcWRKY-18	Chr2	Segmental	0.5320	NaN	NaN
RcWRKY-53	Chr6				
RcWRKY-23	Chr2	Segmental	0.4085	1.6088	0.2539
RcWRKY-58	Chr7				
RcWRKY-24	Chr2	Segmental	0.3239	NaN	NaN
RcWRKY-60	Chr7				
RcWRKY-34	Chr3	Segmental	0.4683	NaN	NaN
RcWRKY-36	Chr4				
RcWRKY-29	Chr3	Segmental	0.3373	1.2935	0.2607
RcWRKY-48	Chr5				
RcWRKY-26	Chr3	Segmental	0.3231	1.3243	0.2440
RcWRKY-52	Chr6				
RcWRKY-37	Chr4	Segmental	0.3757	2.0867	0.1801
RcWRKY-60	Chr7				
RcWRKY-40	Chr4	Segmental	0.4092	1.6738	0.2445
RcWRKY-58	Chr7				
RcWRKY-51	Chr5	Segmental	0.1341	0.2643	0.5075
RcWRKY-50	Chr5				
RcWRKY-41	Chr5	Segmental	0.3781	2.2186	0.1704
RcWRKY-45	Chr5				

## Data Availability

The original contributions presented in the study are included in the article/Appendix A, further inquiries can be directed to the corresponding authors.

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
