# Peer review of "Genome-Wide Identification of *WRKY* Transcription Factor Family in Chinese Rose and Response to Drought, Heat, and Salt Stress"

_genes, 2024, doi:10.3390/genes15060800_

Round 1
Reviewer 1 Report
Comments and Suggestions for Authors
Yan et al
Genome-wide identification of the WRKY transcription factor 1 family in Chinese Rose and response to drought, heat, and salt stress
Review
Introduction
Overall, the Introduction is clear and well written, and justifies the value of the study.
It would probably be best if the genome assembly of the R. chinensis genome used for the study were described and cited here, with a sentence or two such as, “The genome of the diploid R. chinensis has been sequenced and assembled at the chromosomal level. (citation, presumably Raymond et al, as noted below in comments on Section 3.1) The genome and gene annotations are publicly available on the Gramene database web site (http://gramene.org).”
Results
For the benefit of readers who may not wish to read the Methods section in detail, it would be helpful at the beginning of the Results section to explain how the 68 putative WRKY genes in Chinese rose were identified from genomic sequences in databases. For example, something like the following, assuming that this is in fact what was done (hard to be sure from the lack of details in the Methods), “To identify WRKY domain-containing genes, two approaches were used. First, a set of WRKY-containing genes annotated in the Arabidopsis thaliana genome were used to BLAST the set of all annotated genes of the Chinese rose obtained from the Gramene plant genome database annotations of R. chinensis (with literature citation for genome assembly, see below on this point). Second, a consensus WRKY domain model was used independently to search all Chinese rose annotated genes. Candidate hits from these two searches were further analyzed individually, and any likely false positive hits were removed. For details see Methods.”
It is worth noting in the Results that A. thaliana and R. chinensis are both in the Rosid clade of plants, thus are moderately related. That provides justification for using A. thaliana genes as probes to BLAST the R. chinensis genome for putative WRKY genes, rather than using another plant, or else just the WRKY domain consensus model from PFAM.
There is a problem with the numbering of Figures 2 and 3 in the text versus the figures. In Section 2.2, line 136, it says the phylogenetic tree of genes in Arabidopsis and Rosa is Figure 3, but it appears to be Figure 2 according to the figure and the figure legend.
In section 2.3, line 144 says the 68 genes can be grouped into 3 groups and 5 subgroups, shown in Figure 2A. There is no Figure 2A, only a figure 2, which shows the three groups for the genes in both plant species, not just the 68 genes of Rc. Perhaps this text is referring to Figure 3A? However that subpanel does not show any groups or subgroups, just a tree of the 68 genes. The figure should indicate where the 3 groups and 5 subgroups are, if that is the figure that is meant in the text.
Perhaps the order of figures 2 and 3 got switched at some point in the manuscript writing, but the numbering is not currently consistent between the actual Figures/legends and the text.
There seems to be no reference in the text specifically for Figure 3 subpanels B-D (assuming that the text on line 144 is referring to Figure 3A not 2A). Each panel of the figures needs to be mentioned in the text, that is standard paper formatting.
It is not clear what the difference is between a “motif” and a “domain”, please clarify this. The two terms are sometimes used interchangeably, if this is technical jargon coming from particular bioinformatics tools, that should be mentioned in the Methods and also briefly in the Results, otherwise readers may be confused about the difference between figure 3B and 3C.
There seems to be a typo in the Figure 3 legend, line 179, where it mentions 82 RcWRKY genes, whereas there are only 68. Maybe this was text from an earlier version that did not get corrected, but the number of genes needs to be consistent throughout the manuscript.
Section 2.7, protein interaction network, seems completely based on bioinformatic predictions so is of limited utility. It’s a short section so it could stay, since the Discussion describes tests for at least one such interaction prediction in another organism/system (lines 445-447).
Where are the results for the analysis of Gene expression, Section 2.8? There appear to be no figures or tables showing these actual results, whereas they are some of the most interesting as they are experimental and not just bioinformatic predictions and lists. There needs to be some figure or table with the experimental data on the gene expression profiling (suitably interpreted).
According to the journal Genes web site, Supplementary files are allowed, to be published online not in print. It would be reasonable to move several of the Tables and Figures from the Results to an online Supplement, to save on page space in the printed journal. For length lists of genes and primers, most readers would not be interested in seeing the complete lists in the print version, while readers with greater interest could easily get them from an online supplement. Specifically, Tables 1 and 3 should be moved to a Supplement. Table 2 is also long, but contains more specific results that could stay in the main document. Likewise Figure 1 could go in an online Supplement, as the exact chromosomal locations of all 68 genes are not that important to the interesting conclusions of the study.
Methods
Section 3.1, be more specific as to the files that were accessed. Entering Rosa chinensis into the main search tool of the Gramene database yields a very large number of entries, which seem to be all specific gene annotations. Somewhere in the paper there should be a literature citation for the assembly of the R. chinensis genome used for the analysis. The following link was found on the Gramene web site that appears to be for the whole genome assembly, https://ensembl.gramene.org/Rosa_chinensis/Info/Index, which led to this link with more details on the assembly https://ensembl.gramene.org/Rosa_chinensis/Info/Annotation/#assembly, which cites this paper by Raymond et al, https://europepmc.org/article/PMC/5984618 PMID: 29713014 . Confirm whether this the version of the genome used in this study. Add more detail on how the Gramene database was searched, especially the BLAST stringency parameters since results of BLAST searches can be very sensitive to these parameters. A. thaliana and R. chinensis are both Rosids, but after that they are in different Orders, so quite a bit of divergence is expected even at the amino acid level, for homologous genes.
Similarly, be more specific which WRKY protein sequences were obtained from the NCBI database for searches. Searching NCBI Gene with the term “At-WRKY” from lines 261-262 yields no hits, so it is not clear what this term meansFrom the Introduction, it seems that At probably means Arabidopsis thaliana, but this should be clarified in this Methods section 3.1. Searching NCBI Gene with the terms “WRKY” and “Arabidopsis” yields 201 entries, were all these used to generate the gene model for searching the Chinese rose genome annotations? Some seem to be potential pseudogenes or weak gene predictions. It’s not necessary to make a table of 200 or so probe genes if that many were used, but some additional description of the search gene set is needed.
Similarly, the exact WRKY gene model used from PFAM should be named, it’s not clear from the text what model was used. A quick search of the InterPro web site (which has incorporated PFAM) with the term WRKY yields 34 entries, which of these was used to search the Chinese tea genes?
In the genome sequencing paper cited on Gramene, Raymond et al, it states that R. chinensis is a diploid. This is obviously relevant for interpretation of the set of putative WRKY genes, thus it should be mentioned somewhere that the genome under study is diploid, hence different WRKY genes are not just sub-genomic homologue pairs. Since many plants, especially domesticated ones, are polyploid, this point is important to mention in all reports of genomic analyses.
For the stress treatments described in Section 3.5, this reviewer is unfamiliar with the system and cannot comment on whether these treatments were the appropriate ones. The authors appear to know what they are doing, there is no special concern, just to note.
In Section 3.6, clarify the nature of the primers used for first strand cDNA synthesis. Are they random primers? A reader familiar with the particular kit used would probably know this, but for others it is important to note. Given that a fairly small number of genes are being analyzed, it could be imagined that the first strand cDNA synthesis was done using a library of gene-specific primers. Just clarify this minor point.
For the quantitative gene expression analysis, some more detail is needed, it is not sufficient just to say in lines 344-245, “the 2-DDCt method was used”. How were the raw qPCR data normalized, with what set of control genes? This is important to include since some differential expression is reported in the Results, in order for readers to critically interpret the findings. Also for that same sentence, the part “Data reliability was determined from the amplification and dissolution curves” it not clear at all, in what sense do the qPCR curves provide evidence of data “reliability”? Also what are “dissolution curves”, that is not standard English?
The various other bioinformatic analyses described in the Methods section appear to be appropriate and used correctly, although this reviewer is not familiar with all the various tools (there are so many these days).
Discussion
As a minor formatting point, probably easily fixed in typesetting, on page 16 the Discussion starts side by side with Table 3, and was not easy to find. The Discussion section should start after the Table in the formatted final version of the document.
The Discussion in general is well written and clear. It is a bit long given that much of the study presents bioinformatic predictions, to be verified with future experimental work. If page space is tight, much of the Discussion could be shortened without loss of the most important points.
Conclusion
Line 503, it is a bit too strong to say that the study “revealed the key functions of RcWRKY genes…” There are no actual functional studies as such, in terms of biochemical studies of DNA binding, or genetic variations. The transcriptomic work is interesting, but it does not demonstrate function, just correlations. It would be more correct to say something like “Bioinformatic and gene expression analyses provide hints as to the functions of some of the identified RcWRKY genes, to be tested with further experimental work.” Or something like that.
Comments on the Quality of English LanguageThe English is overall good, with a few minor issues to be corrected, as mentioned in the detailed comments to authors.
Author Response
Dear reviewer,
Thank you very much for you to send me the reviewers’ comments on my manuscript genes-3008848 entitled "Genome-wide identification of the WRKY transcription factor family in Chinese rose and response to drought, heat, and salt stress". As soon as I received the E-mail, the article has been polished by a professional company. All of the reviewers’ comments are responded point by point. I hope these modifications will make the manuscript qualified to be published in the journal. Thank you very much for your time and effort that goes into the publication of this paper.
Minghua Deng
Introduction
Overall, the Introduction is clear and well written, and justifies the value of the study.
It would probably be best if the genome assembly of the R. chinensis genome used for the study were described and cited here, with a sentence or two such as, “The genome of the diploid R. chinensis has been sequenced and assembled at the chromosomal level. (citation, presumably Raymond et al, as noted below in comments on Section 3.1) The genome and gene annotations are publicly available on the Gramene database web site (http://gramene.org).”
>>The genome assembly of the R. chinensis genome used for the study were described and cited.
Results
For the benefit of readers who may not wish to read the Methods section in detail, it would be helpful at the beginning of the Results section to explain how the 68 putative WRKY genes in Chinese rose were identified from genomic sequences in databases. For example, something like the following, assuming that this is in fact what was done (hard to be sure from the lack of details in the Methods), “To identify WRKY domain-containing genes, two approaches were used. First, a set of WRKY-containing genes annotated in the Arabidopsis thaliana genome were used to BLAST the set of all annotated genes of the Chinese rose obtained from the Gramene plant genome database annotations of R. chinensis (with literature citation for genome assembly, see below on this point). Second, a consensus WRKY domain model was used independently to search all Chinese rose annotated genes. Candidate hits from these two searches were further analyzed individually, and any likely false positive hits were removed. For details see Methods.”
>>It has been explained at the beginning of the Results section.
It is worth noting in the Results that A. thaliana and R. chinensis are both in the Rosid clade of plants, thus are moderately related. That provides justification for using A. thaliana genes as probes to BLAST the R. chinensis genome for putative WRKY genes, rather than using another plant, or else just the WRKY domain consensus model from PFAM.
>>It has been added that Arabidopsis and roses are moderately related, so using A. thaliana genes as probes to BLAST the R. chinensis genome for putative WRKY genes.
There is a problem with the numbering of Figures 2 and 3 in the text versus the figures. In Section 2.2, line 136, it says the phylogenetic tree of genes in Arabidopsis and Rosa is Figure 3, but it appears to be Figure 2 according to the figure and the figure legend.
>>It has been modified to Figure 2.
In section 2.3, line 144 says the 68 genes can be grouped into 3 groups and 5 subgroups, shown in Figure 2A. There is no Figure 2A, only a figure 2, which shows the three groups for the genes in both plant species, not just the 68 genes of Rc. Perhaps this text is referring to Figure 3A? However that subpanel does not show any groups or subgroups, just a tree of the 68 genes. The figure should indicate where the 3 groups and 5 subgroups are, if that is the figure that is meant in the text.
>>This text is referring to Figure 3A, the figure has indicated where the 3 groups are.
Perhaps the order of figures 2 and 3 got switched at some point in the manuscript writing, but the numbering is not currently consistent between the actual Figures/legends and the text.
>>The numbering is currently consistent between the actual Figures/legends and the text.
There seems to be no reference in the text specifically for Figure 3 subpanels B-D (assuming that the text on line 144 is referring to Figure 3A not 2A). Each panel of the figures needs to be mentioned in the text, that is standard paper formatting.
>>Figure 3 subpanels B-D have been mentioned in the text.
It is not clear what the difference is between a “motif” and a “domain”, please clarify this. The two terms are sometimes used interchangeably, if this is technical jargon coming from particular bioinformatics tools, that should be mentioned in the Methods and also briefly in the Results, otherwise readers may be confused about the difference between figure 3B and 3C.
>>the difference between a “motif” and a “domain” has been mentioned in the Methods and also briefly in the Results.
There seems to be a typo in the Figure 3 legend, line 179, where it mentions 82 RcWRKY genes, whereas there are only 68. Maybe this was text from an earlier version that did not get corrected, but the number of genes needs to be consistent throughout the manuscript.
>>It has been revised.
Section 2.7, protein interaction network, seems completely based on bioinformatic predictions so is of limited utility. It’s a short section so it could stay, since the Discussion describes tests for at least one such interaction prediction in another organism/system (lines 445-447).
>>OK.
Where are the results for the analysis of Gene expression, Section 2.8? There appear to be no figures or tables showing these actual results, whereas they are some of the most interesting as they are experimental and not just bioinformatic predictions and lists. There needs to be some figure or table with the experimental data on the gene expression profiling (suitably interpreted).
>>Sorry, I forgot to include these figure when organizing and submitting them, but they have already been added.
According to the journal Genes web site, Supplementary files are allowed, to be published online not in print. It would be reasonable to move several of the Tables and Figures from the Results to an online Supplement, to save on page space in the printed journal. For length lists of genes and primers, most readers would not be interested in seeing the complete lists in the print version, while readers with greater interest could easily get them from an online supplement. Specifically, Tables 1 and 3 should be moved to a Supplement. Table 2 is also long, but contains more specific results that could stay in the main document. Likewise Figure 1 could go in an online Supplement, as the exact chromosomal locations of all 68 genes are not that important to the interesting conclusions of the study.
>>Tables 1 and 3 and Table 2 will be moved to a Supplement.
Methods
Section 3.1, be more specific as to the files that were accessed. Entering Rosa chinensis into the main search tool of the Gramene database yields a very large number of entries, which seem to be all specific gene annotations. Somewhere in the paper there should be a literature citation for the assembly of the R. chinensis genome used for the analysis. The following link was found on the Gramene web site that appears to be for the whole genome assembly, https://ensembl.gramene.org/Rosa_chinensis/Info/Index, which led to this link with more details on the assembly https://ensembl.gramene.org/Rosa_chinensis/Info/Annotation/#assembly, which cites this paper by Raymond et al, https://europepmc.org/article/PMC/5984618 PMID: 29713014 . Confirm whether this the version of the genome used in this study. Add more detail on how the Gramene database was searched, especially the BLAST stringency parameters since results of BLAST searches can be very sensitive to these parameters. A. thaliana and R. chinensis are both Rosids, but after that they are in different Orders, so quite a bit of divergence is expected even at the amino acid level, for homologous genes.
>>The files that were accessed have been added,After confirmation, the ensamble plant database is being used,the link has been added,and the BLAST stringency parameters have been added.
Similarly, be more specific which WRKY protein sequences were obtained from the NCBI database for searches. Searching NCBI Gene with the term “At-WRKY” from lines 261-262 yields no hits, so it is not clear what this term meansFrom the Introduction, it seems that At probably means Arabidopsis thaliana, but this should be clarified in this Methods section 3.1. Searching NCBI Gene with the terms “WRKY” and “Arabidopsis” yields 201 entries, were all these used to generate the gene model for searching the Chinese rose genome annotations? Some seem to be potential pseudogenes or weak gene predictions. It’s not necessary to make a table of 200 or so probe genes if that many were used, but some additional description of the search gene set is needed.
>>After confirmation, the TAIR database is being used to get one WRKY protein sequence,AtWRKY4(AT1G13960.1)was used,and TBtools was used for BLAST.
Similarly, the exact WRKY gene model used from PFAM should be named, it’s not clear from the text what model was used. A quick search of the InterPro web site (which has incorporated PFAM) with the term WRKY yields 34 entries, which of these was used to search the Chinese tea genes?
>>(PF03106) was used to search the Chinese rose genes.
In the genome sequencing paper cited on Gramene, Raymond et al, it states that R. chinensis is a diploid. This is obviously relevant for interpretation of the set of putative WRKY genes, thus it should be mentioned somewhere that the genome under study is diploid, hence different WRKY genes are not just sub-genomic homologue pairs. Since many plants, especially domesticated ones, are polyploid, this point is important to mention in all reports of genomic analyses.
>> It has been mentioned in the discussion section.
For the stress treatments described in Section 3.5, this reviewer is unfamiliar with the system and cannot comment on whether these treatments were the appropriate ones. The authors appear to know what they are doing, there is no special concern, just to note.
>>OK.
In Section 3.6, clarify the nature of the primers used for first strand cDNA synthesis. Are they random primers? A reader familiar with the particular kit used would probably know this, but for others it is important to note. Given that a fairly small number of genes are being analyzed, it could be imagined that the first strand cDNA synthesis was done using a library of gene-specific primers. Just clarify this minor point.
>>This issue has been clarified.
For the quantitative gene expression analysis, some more detail is needed, it is not sufficient just to say in lines 344-245, “the 2-DDCt method was used”. How were the raw qPCR data normalized, with what set of control genes? This is important to include since some differential expression is reported in the Results, in order for readers to critically interpret the findings. Also for that same sentence, the part “Data reliability was determined from the amplification and dissolution curves” it not clear at all, in what sense do the qPCR curves provide evidence of data “reliability”? Also what are “dissolution curves”, that is not standard English?
>>Control gene has been added,and “dissolution curves” has been revised.
The various other bioinformatic analyses described in the Methods section appear to be appropriate and used correctly, although this reviewer is not familiar with all the various tools (there are so many these days).
>>ok.
Discussion
As a minor formatting point, probably easily fixed in typesetting, on page 16 the Discussion starts side by side with Table 3, and was not easy to find. The Discussion section should start after the Table in the formatted final version of the document.
>>ok.
The Discussion in general is well written and clear. It is a bit long given that much of the study presents bioinformatic predictions, to be verified with future experimental work. If page space is tight, much of the Discussion could be shortened without loss of the most important points.
>>ok.
Conclusion
Line 503, it is a bit too strong to say that the study “revealed the key functions of RcWRKY genes…” There are no actual functional studies as such, in terms of biochemical studies of DNA binding, or genetic variations. The transcriptomic work is interesting, but it does not demonstrate function, just correlations. It would be more correct to say something like “Bioinformatic and gene expression analyses provide hints as to the functions of some of the identified RcWRKY genes, to be tested with further experimental work.” Or something like that.
>>It has been revised.
Reviewer 2 Report
Comments and Suggestions for Authors
Please see attached

Minor proof reading
Author Response
Dear reviewer,
Thank you very much for you to send me the reviewers’ comments on my manuscript genes-3008848 entitled "Genome-wide identification of the WRKY transcription factor family in Chinese rose and response to drought, heat, and salt stress". As soon as I received the E-mail, the article has been polished by a professional company. All of the reviewers’ comments are responded point by point. I hope these modifications will make the manuscript qualified to be published in the journal. Thank you very much for your time and effort that goes into the publication of this paper.
Minghua Deng
In this exiting study about the genome-wide identifcation of the WRKY transcription factors in Chinese Rose, sound bioinformatics as well as detailed writing and good analysis were conducted. The manuscript may be further improved with following suggestions:
Page 1-2 and Fgure legends - please check journal's formatting requirements
>>It has been modified.
Line 58 - space missing after 'chrysanthemum'
>>It has been modified.
Line 80 - double space before [11]
>>It has been modified.
Line 111 - please specify how they were identifed (sequence alignments, database search, etc.)
>>It has been modified.
Line 218 - legend of the Fgure 6 is missing.
>>It has been modified.
Line 227 - legend of the Fgure 7 is missing.
>>It has been modified.
Line 315-316 - the font is not the same as in the whole manuscript.
>>It has been modified.
Line 510-511 - suggest to add importance if this plant for agriculture.
>>It has been modified.